# Reduction in the Cocoa Spontaneous and Starter Culture Fermentation Time Based on the Antioxidant Profile Characterization

**DOI:** 10.3390/foods12173291

**Published:** 2023-09-01

**Authors:** César R. Balcázar-Zumaeta, Alexa J. Pajuelo-Muñoz, Deisy F. Trigoso-Rojas, Angel F. Iliquin-Chavez, Editha Fernández-Romero, Ives Yoplac, Lucas D. Muñoz-Astecker, Nadia Rodríguez-Hamamura, Ily M. Maza Mejía, Ilse S. Cayo-Colca, Gilson C. A. Chagas-Junior, Jorge L. Maicelo-Quintana, Efrain M. Castro-Alayo

**Affiliations:** 1Instituto de Investigación, Innovación y Desarrollo para el Sector Agrario y Agroindustrial (IIDAA), Universidad Nacional Toribio Rodríguez de Mendoza de Amazonas, Chachapoyas 01001, Peru; 7255974661@untrm.edu.pe (A.J.P.-M.); 7193222461@untrm.edu.pe (D.F.T.-R.); 7352940072@untrm.edu.pe (A.F.I.-C.); editha.fernandez@untrm.edu.pe (E.F.-R.); lucas.munoz@untrm.edu.pe (L.D.M.-A.); efrain.castro@untrm.edu.pe (E.M.C.-A.); 2Programa de Doctorado en Ciencias Agrarias, Escuela de Posgrado, Universidad Nacional de Piura, Jr. Tacna 748, Piura 20002, Peru; 3Facultad de Ingeniería Zootecnista, Agronegocios y Biotecnología, Universidad Nacional Toribio Ro-Dríguez de Mendoza de Amazonas, Chachapoyas 01001, Peru; ives.yoplac@untrm.edu.pe (I.Y.); icayo.fizab@untrm.edu.pe (I.S.C.-C.); jmaicelo@untrm.edu.pe (J.L.M.-Q.); 4Laboratorio LABICER, Facultad de Ciencias, Universidad Nacional de Ingeniería, Av. Tupac Amaru 210, P.O. Box 15000, Rímac 15333, Peru; nrodriguezh@uni.edu.pe; 5Laboratorio de Investigación de Química Analítica y Ambiental, Universidad Nacional de Ingeniería, Av. Tupac Amaru 210, P.O. Box 15000, Rímac 15333, Peru; imaza@uni.edu.pe; 6Laboratório de Processos Biotecnológicos (LABIOTEC), Programa de Pós Graduação em Ciência e Tecnologia de Alimentos (PPGCTA), Instituto de Tecnologia (ITEC), Universidade Federal do Pará (UFPA), Rua Augusto Corrêa, 01, Campus Guamá, Belém 66075-110, Brazil; chagasjunior.gca@gmail.com

**Keywords:** antioxidants, aromatics compounds, Criollo cocoa, chocolate, fermentation stages, polyphenols

## Abstract

In current systems, the fermentation spontaneous process produces fermented beans of heterogeneous quality due to the fermentation time. This study demonstrated that the fermentation time should be reduced. For this purpose, the physicochemical parameters, antioxidant profile, and volatile compounds were characterized in two types of fermentation (spontaneous and starter culture) for 168 h in cocoa from three altitude levels. Multivariate analysis (cluster and PCA) was used to discriminate the fermentation stages. We found three stages in all fermentations, where the first two stages (0 h to 96 h) were characterized by a higher antioxidant potential of the cocoa bean and the presence of desirable volatile compounds such as acids, alcohols, aldehydes, ketones, and esters, which are precursors of cocoa aroma; however, prolonged fermentation times affected the antioxidant profile of the bean. In addition, the use of a starter culture facilitates the release of compounds in a shorter time (especially alcohols and esters). It is concluded that it is necessary to reduce the fermentation time under these conditions in the region of Amazonas.

## 1. Introduction

Cocoa (*Theobroma cacao* L.) is a cash crop associated with various nutritional benefits and high antioxidant activity [1]. Among the genetic varieties, Criollo cocoa stands out for its bioactive and sensory quality in final products such as chocolate [2]. In the case of the region of Amazonas (Peru), Criollo cocoa is a variety known as native, whose denomination of origin is “Cacao Amazonas Peru”, produced only in the provinces of Bagua and Utcubamba [3].

To achieve the complex formation of the characteristic flavor of the cocoa bean, a crucial process is fermentation [4,5], during which the pulp is removed from the bean [6]. In addition, several biochemical, enzymatic, and microbiological reactions occur in this process, resulting in volatile fractions and metabolites that are precursors of cocoa flavor and aroma [7], which later, during drying and roasting, lead to the presence of chemical compounds that determine an adequate chemical and sensory quality of chocolate [8]. Currently, this process has a very heterogeneous duration, depending, among other things, on the variety [9]. According to Balcázar-Zumaeta et al. [10] and Castro-Alayo et al. [3], fermentation lasts between 4 and 7 days and can affect the concentration of metabolites, hence the sensory quality. This process is called spontaneous fermentation [11,12] and produces unpleasant flavors if it is subjected to prolonged periods due to an increase in bacilli and filamentous fungi [10].

Balcázar-Zumaeta et al. [10] have reported several studies that, through multivariate analysis based on key biomarkers (non-volatile and volatile), allow distinguishing three stages of cocoa fermentation. This can help to identify an ideal fermentation time [13], for which interest has arisen in shorter fermentations (especially in Criollo cocoa), as it improves the delicate flavor of the bean by reducing bitterness and astringency [12,14].

An alternative that has emerged to minimize quality variation in fermented beans is the addition of starter cultures to the fermentation system [15]. Here, potential microorganisms synthesize flavor compounds and their derivatives during fermentation [16], which reduces the time and improves the concentrations to obtain a controlled fermentation of cocoa within this group. Some reported microorganisms are yeasts (*Kluyveromyces marxianus*, *Pichia kluyveri*, and *Saccharomyces cerevisiae*), lactic acid bacteria (*Lactobacillus fermentum* and *L. plantarum*), and acetic acid bacteria (*Acetobacter pasteurianus*, *A. aceti*, and *A. tropicalis*) [17].

There are publications related to the characterization of stages during cocoa fermentation in different varieties [4,7,18,19] and countries [8,12,15,20,21]. Studies agree that the fermentation time is a key factor for cocoa quality [22]; however, the reported information is very scarce for the case of Peru, especially regarding the use of starter cultures and stages during bean fermentation. In the study of Castro-Alayo et al. [23], two stages of fermentation based on the crystallization and polymorphism of the cocoa butter (CB) were found. They proposed that a three-day fermentation is recommended for an acceptable crystallization.

Since no further research has been reported, there is a need to modulate cocoa bean quality, which requires characterizing fermentation times in terms of key compounds [24] and monitoring physicochemical parameters and aromatic biomarkers. Therefore, here, we report for the first time the reduction in spontaneous (SF) and starter culture (*S. cerevisiae*, SC) fermentation time based on the antioxidant profile characterization of cocoa beans harvested in Amazonas, Peru.

## 2. Materials and Methods

### 2.1. Biological Material and Reagents

The Criollo (native) cocoa beans were collected from the populated centers of Guadalupe (5°33′45.14″ S, 78°32′50.53″ W, 420 mamsl), Tolopampa (5°39′21.25″ S, 78°29′40.44″ W, 514 mamsl), and Copallin (5°41′13.70″ S, 78°24′20.58″ W, 869 mamsl) in the province of Bagua, Amazonas region (see Figure 1). All data were collected in situ using a Global Positioning System (Garmin Montana^®^ 680; Garmin Ltd., Olathe, KS, USA).

Reagents used: Folin–Ciocalteu phenol reagent (Merck, Darmstadt, Germany), sodium carbonate (Spectrum, New Brunswick, NJ, USA), *Saccharomyces cerevisiae* derived from ATCC^®^ 18824™ (01066K, Microbiologics, Saint Cloud, MN, USA), Methanol HPLC grade (JT Baker, Deventer, The Netherlands). In the case of: hydrochloric acid (HCl), petroleum ether, potassium persulfate, gallic acid, ethanol, acetonitrile, YPD agar, acetic acid, peptone water, HPLC standards (caffeine, Theobromine, (+)-catechin, (−)-epicatechin), (±)-6-hydroxy-2,5,7,8-tetramethylchromane-2-carboxylic acid (Trolox), 2,2′-azino-bis (3-ethylbenzothiazoline-6-sulfonic acid) diammonium salt (ABTS), 2,2-Diphenyl-1-picrylhydrazyl (DPPH), Fe^3+^-TPZ, butanol, and acetate buffer hydrate were purchased from Sigma Aldrich (St. Louis, MO, USA).

### 2.2. Yeast Strain (Saccharomyces cerevisiae) Activation

The activation process of *S. cerevisiae* (derived from ATCC^®^18824^TM^, Saint Cloud, MN, USA) was carried out according to the method described by Chagas et al. [20]. The strains were activated in Petri dishes containing sterile YPD (Yeast Extract Peptone Dextrose) agar; these were incubated at 28 °C for 72 h. Colonies from the culture were transferred to new plates using peptonized water in a liquid medium. To ferment 40 kg of cocoa, the culture was prepared in an Erlenmeyer flask with 500 mL of peptonized water; then, 16 plates with the activated strains were used, adding 2 mL of peptonized water to each plate. Then, the contents of each plate were transferred to the flask with peptonized water (cell growth ≈ 10^8^ cells/mL). Finally, the media with the strains were transported to the APROCAM cooperative to be added to the cocoa fermentation processes.

### 2.3. Criollo Cocoa Fermentation

The beans described in Figure 1 were of the Criollo (native) variety, harvested, transported, and processed in the Cooperativa de Servicios Múltiples APROCAM, located at Car. Bagua- Copallín Km.4—Province of Bagua, for processing.

Two types of fermentation, spontaneous (SF) and starter culture (SC), following the procedure described by Castro-Alayo et al. [23], were carried out. The process was similar in both cases: working in drawers internally covered with stainless steel, 40 kg of fresh cocoa were deposited and covered with polyethylene bags to concentrate at a higher temperature; later, at 48 h, the first turning was performed (changing the drawer), and the same was repeated at 96 h and 144 h. The fermentations lasted a total of 168 h (one week). In the case of the fermentation with SC, the cocoa beans were mixed with ≈10^6^ cells of *S. cerevisiae*/cocoa grams before starting the fermentation.

### 2.4. Fermentation Monitoring (SF and SC) and Sampling

Considering the works of Castro-Alayo et al. [23], Chagas et al. [20], and Velásquez-Reyes et al. [25], during the fermentation processes (SF and SC), the parameters of temperature, pH, and dissolved oxygen were recorded every 24 h using a multiparameter (Lutron, WA-2017SD, Coopersburg, PA, USA). In terms of sample collection, 150 g of cocoa beans (previously separated from the pulp) was extracted daily, placed in sterile bags, and subjected to a thermal shock with liquid nitrogen from the beginning of fermentation (0 h) to the end of fermentation (168 h) to preserve the conditions at each sampling time. The samples were transferred to the Food and Postharvest Engineering Research Laboratory of UNTRM for preservation under ultra-freezing conditions (Eppendorf, Premium 0410, Hamburg, Germany) until further analysis.

### 2.5. Physicochemical Parameters of Fermented Native Cocoa Beans

#### 2.5.1. Water Activity and Moisture of Cocoa Beans

The water activity and moisture of cocoa beans was determined as described by Castro-Alayo et al. [23] using a portable water analyzer (Rotronic AG, HygroPalm, Bassersdorf, Switzerland). For this purpose, beans were placed without filling the container, then the probe was connected, and after 5 min, the Aw was recorded. Moisture was measured using a halogen light moisture analyzer (Mettler Toledo, Excellence Plus HX204, Greifensee, Switzerland) at 105 °C until the moisture content (%MC) was obtained.

#### 2.5.2. Titratable Acidity and pH of Cocoa Beans

These were determined according to AOAC methods 942.15 and 970.21 [26].

#### 2.5.3. Fermentation Index (FI)

Following Hinneh et al. [27] with modifications, the index was determined by mixing cocoa (0.1 g) with methanol: HCl solution (97:3 *v*/*v*), which was vortexed (Analog Mini Vortex Mixers, OHAUS, Pine Brook, NJ, USA) for 1 min. It was stored at 6 °C for 19 h and then filtered and adjusted with 10 mL of methanol: HCl (97:3 *v*/*v*). Finally, the absorbances at 460 nm and 530 nm were measured in a UV-Vis spectrophotometer (EMCLAB, EMC-11-UV, Duisburg, Germany). The IF was estimated as the ratio of the absorbance readings at 460 nm and 530 nm.

### 2.6. Cocoa Freeze Drying and Defatting

For the subsequent analyses, the collected samples were freeze-dried (Labconco, 710402010 model, Kansas City, MO, USA); inside the containers, the bean samples were placed in falcon tubes (50 mL) at 0.008 bar and −84 °C for 18 h. After obtaining freeze-dried cocoa beans, the samples were defatted according to the work of Hernández-Hernández et al. [28] in a Soxhlet equipment (Daihan Scientific, Seoul, Republic of Korea) using petroleum ether as a solvent, controlling eight siphoning. The defatted samples were exposed to the environment to evaporate the residues of the solvent and were stored in a dry place.

### 2.7. Antioxidant Profile of Native Cocoa during Fermentation

For the evaluations, extracts were prepared as described by Melo et al. [29], where 1 g of defatted cocoa was used in 50 mL of methanol (80%), mixed in a vortex shaker for 1 min, then centrifuged (Eppendorf, 5810R, Hamburg, Germany) (12 rpm for 10 min) and passed through a Millipore filter (0.45 μm) (Millex, Merck, Darmstadt, Germany) into vials that were stored at −6 °C until further use.

#### 2.7.1. Total Polyphenol Content

The TPC was determined by the Folin–Ciocalteu method, using a UV-Vis spectrophotometer (SECOMAM, Uv Line 9400, Alès, France) at an absorbance of 750 nm [30], the content was expressed as mg gallic acid (GAE) per gram of sample (a gallic acid calibration curve was used, y = 0.2213x − 0.1924, R^2^ = 0.9933).

#### 2.7.2. Total Anthocyanin Content

The differential pH method was used, according to Mihai et al. [31], with some modifications. A mixture of 100 μL of cocoa extract and 900 μL of buffers (NaCl pH 1.0 and C_2_H_3_NaO_2_ pH 4.5) was made, and readings were performed using a UV-Vis spectrophotometer (GENESYS^TM^, Thermo Scientific, Waltham, MA, USA) at 530 nm and 700 nm. The contents were expressed in mg cyanidin-3-glucoside/100 g of sample.

#### 2.7.3. Antioxidant Activity

The following methods were used to determine this activity:DPPH method: The method described by Gültekin-Özgüven et al. [1] was used based on the reduction capacity of the DPPH radical (2,2-Diphenyl-1-Picrylhydrazyl); for this purpose, cocoa extract (100 μL) and DPPH solution (3.9 mL) were used. The absorbances were measured in a UV-Vis spectrophotometer (SECOMAM, Uv Line 9400, Alès, France) at an absorbance of 517 nm. A standard curve was performed using Trolox (y = −0.0004x + 0.8502, R^2^ = 0.9993) to determine the antioxidant activity. Finally, the results were expressed in µmol TE/g sample.ABTS method: This method is based on ABTS (2,2′-azino-bis (3-ethylbenzothiazoline-6-sulfonic acid)) free radical scavenging activity, following Floegel et al. [32] and Godočiková et al. [33]. The radical decolorization assay was performed by measuring the absorbance at 734 nm using a UV-Vis spectrophotometer (SECOMAM, Uv Line 9400, Alès, France). A standard curve was performed using Trolox (y = −0.0004x + 0.8502, R^2^ = 0.9993) to determine the antioxidant activity. Finally, the results were expressed in µmol TE/g sample.FRAP method: According to Mihai et al. [31], it is a colorimetric method linked to ferric ion reduction (Fe^3+^-TPTZ). The FRAP reagent was mixed with acetate buffer (pH 3.6), TPTZ (0.1 M) diluted in hydrochloric acid (0.4 M), and ferric chloride hexahydrate (0.2 M). The mixture was pipetted; then, the extract (90 µL) and distilled water (260 μL) were added; next, the sample was taken to a water bath (Indumelab, BM-10, Lima, Peru) at 30 °C for 4 min, and finally, reading at 593 in a UV-Vis Vis spectrophotometer (EMCLAB, EMC-11-UV, Duisburg, Germany). A standard curve was performed to determine the antioxidant activity (y = 0.0006x + 0.0853, R^2^ = 0.9949). The results were expressed as μmol Fe^2+^/100 g sample.

#### 2.7.4. Quantification of Catechin, Epicatechin, Caffeine, and Theobromine

Separation and quantification was performed by ultra-high-performance liquid chromatography (UHPLC, Agilent Technologies, 1290 Infinity II, Waldbronn, Germany) with another multisample (G7167B), flexible pump (G7104A), column oven (G7116B) and diode array detector (G7117B). The method described by Balcázar-Zumaeta et al. [30] was used, for which the extracts (10 μL) were filtered (0.45 μm) using a syringe (Millex, Merck, Darmstadt, Germany). Separation was performed with a C18 column (100 mm × 4.6 mm O.D.S.-2, 3 μm). The following phases were used: (A): 2% acetic acid in water, (B): mixture of acetonitrile, water, and acetic acid (400:90:10 *v*/*v*/*v*/*v*). The flow rate was 0.75 mL/min by linear gradient elution: 10–14% B (5 min), 14–23% B (11 min), 23–35% B (5 min), 35–40% B (14 min), 40–100% B (3 min), 100% B isocratic (3 min), 100–10% B (3 min) and 10% B isocratic (4 min).

UV performed detection at a wavelength of 280 nm (each run was at 26 °C for 20 min), and the column was at a temperature of 40 °C. Quantifying the four metabolites was performed by comparison with the areas of the peaks of each standard, using ChemStation control software (version A.02.14 05-16 for OpenLAB). The results were expressed as mg of the metabolite/g sample.

### 2.8. Aromatic Profile in Native Cocoa Beans during Fermentation

#### 2.8.1. Extraction of Volatile Compounds

The procedure described by Valle-Epquín et al. [34] was followed with some modifications. Five grams of cocoa fermented beans was fragmented in a porcelain mortar and deposited into a 10 mL vials (PTFE/Silicone septa) plus 3 mL of ultrapure water and 50 μL of butanol (HPLC grade). The vials were hermetically closed with a septum and left until the equilibrium time was reached. The extraction of volatiles compounds was performed using divinylbenzene/carboxene/polydimethylsiloxane (DVB/CAR/PDMS) fibers with an outer diameter of 50/30 µm. The fiber was conditioned in the chromatograph injector (250 °C for 5 min); then, it was exposed to sample/headspace (50 °C for 45 min).

#### 2.8.2. Gas Chromatograph Conditions

A gas chromatograph (Shimadzu Corp., GC-2010 Plus, Kyoto, Japan) with an Rtx-Wax capillary column (30 m × 0.25 mm ID × 0.25 µm df, Restek) was used. For the work, the oven was programmed at 40 °C for 5 min, then increased to 140 °C at a flow rate of 2 °C/min, and finally increased to 250 °C at a flow rate of 10 °C/min for 66 min; in addition, He was worked with He as carrier gas (0.75 mL/min). The injector (splitless mode) and the detector were conditioned at 250 °C with an SPME fiber desorption of 5 min [35].

#### 2.8.3. Identification of Volatile Compounds

A mass spectrometer (Shimadzu Corp., GCMS-QP2010 Ultra, Kyoto, Japan) was used to identify the spectra at 70 eV (30 to 500 amu (*m*/*z*) with a scan speed of 1666 and an event time of 0.3 s. Source T: 230 °C, interphase: 250 °C [35]. The spectra recording during elution was automatic using GCMS Real Time Analysis software (version 4.30, Shimadzu Corp.). And the compounds having aroma descriptors were identified using the Flavor and Extract Manufacturers Association (FEMA) online database [34].

#### 2.8.4. Quantification of Volatile Compounds

A comparison was made with the National Institute of Standards and Technology-NIST library database for this purpose. For the content of each volatile compound, the equation described by Escobar et al. [13] was used:(1)$$q_{i}(\mu g/g)=\frac{25\times A_{i}}{A_{but}\times m_{e}\times W},$$
where

*q_i_*: quantity of compound i;

*A_i_*: area of compound i;

*A_but_*: area of butanol (standard);

25: butanol content in mgkg;

*m_e_*: mass of sample introduced into the vial in g;

*W*: water contained in the sample.

### 2.9. Data Analysis

The data were analyzed in triplicate for the beans’ physicochemical characterization and the cocoa beans’ antioxidant profile. The volatile profile was analyzed only once at each time. The results were subjected to analysis of variance (ANOVA) for each type of fermentation and cocoa origin, and then, a multiple comparisons test (Tukey, 95%) was performed. To identify the stages in cocoa fermentation, multivariate analysis was applied, including the variables of the antioxidant profile (AA, TPC, AC, flavan-3-ols, and methylxanthines), for which a cluster analysis (k-means) and principal component analysis (PCA) were performed. Statistical analyses were performed using RMarkdown free software (RStudio, version 2022.07.2+576, Boston, MA, USA).

## 3. Results and Discussion

### 3.1. Fermentation Processes Monitoring (SF and SC) in Cocoa Pulp-Bean Mass

Figure 2 shows the temperature profiles in the middle of the boxes for both types of fermentation. In the first hours (0 to 72 h), the temperature of the fermenting cocoa pulp-bean mass was below 40 °C, with a relatively constant pH, similar to that reported by Melo et al. [29]. The temperature also increased from 25 °C to about 48 °C in both types of fermentation (Appendix A), similar to the range reported by Racine et al. (2019), which was between 25 and 55 °C, associated with the release of heat trapped by CO_2_ in the fermentation box [11,36] due to ethanol metabolism by BAA and the conversion of the available substrate (sugar) [4,29], leading to phenolic degradation (Table 1 and Table 2), as proposed by Racine et al. [37]. Likewise, temperatures remained within the expected range and with very similar values, demonstrating that the use of *S. cerevisiae* does not affect the temperature of the fermentative cocoa pulp-bean mass with and without starter culture [38].

Melo et al. [29] report that good fermentation reaches a temperature between 45 and 48 °C at 72 h, which is a temperature similar to ours (cocoa from Copallín and Guadalupe). On the other hand, cocoa from the Tolopampa reached the temperatures above at a time higher than 72 h, which was possibly due to its origin [37]. In the case of the fermentation of the cocoa pulp-bean mass from Tolopampa, a heat loss was detected at the end of the process due to the temperature changes where the process was carried out [39], requiring a longer fermentation time to reach the appropriate temperature.

In our study, the increase in pH in the fermenting cocoa pulp-bean mass could be due to frequent stirring every 48 h, similar to what was found by Portillo and de Farinas [40]. The pH of SF and SC slightly increases at the end of the fermentation in the cocoa pulp-bean mass, in the range of 4.45 to 5.06 (Appendix A), which was accompanied by an increase in temperature similar to those reported by Santander et al. [41]. Furthermore, these changes in pH and temperature (Figure 2) lead to pulp degradation and cotyledon death in cocoa [42,43].

On the other hand, the behavior of dissolved oxygen varied according to the origin of the cocoa; initially, in SF, an OD level of 5.6, 2.73, and 0.36 mg/L was recorded for cocoa from Guadalupe, Copallín, and Tolopampa, respectively, while in SC, the OD at the beginning was 0.76, 2.93 and 0.63 mg/L, respectively. It could be confirmed that at 24 h, the DO level decreased sharply, as reported in the study of Racine et al. [37]. Likewise, the OD during SF and SC presented enormous differences relevant to the different physiological activities presented by cocoa [44]. In the period from 24 to 96 h, an increase in DO was observed, which was accompanied by an increase in the temperature of the fermenting cocoa pulp-bean mass; this behavior is explained by the consumption of oxygen by BAA, causing exothermic oxidation reactions [45,46].

### 3.2. Physicochemical Parameters in Cocoa Beans during SF and SC

In Figure 3, concerning the internal pH in the bean, the values in both fermentations were above 4.3, higher than that reported for the Forastero variety, which was 3.32 [25]. In addition, the pH tended to decrease at the end of the fermentation process (in SF, it reached 4.48, and in SC, it reached 4.34; see Appendix A). The results are similar to those obtained by De Vuyst and Weckx [47], where the decrease was in the range of 4.5–5 (well-fermented beans), and in the study of Calvo et al. [39], who recorded pH values from 4.5 to 4.8, which can be explained by the activity of the fermenting bacteria (lactic and acetobacter) that promote the incorporation of organic acids [36,48,49,50]. Following the behavior of internal pH, Afoakwa et al. [51] and Dewandari et al. [49] consider that a pH close to 5.0 is related to cocoa with better aromatic and sensory profiles; in our study, a pH of 4.9 and 5.5 (Figure 3) was observed in SF between 48 and 96 h, while in SC, it was between 48 and 72 h, indicating a difference in the final cocoa quality [39]; the data are presented in the Appendix A).

Regarding the fermentation index (FI) of the beans in both treatments, it can be seen that it increases with time, similar to that reported by Caporaso et al. [52]. Initially, the IF was below 1 until 48 h in both fermentations and specifically below 1 until 120 h in cocoa fermentation from Copallín. This suggests that environmental factors influence the time to reach an acceptable degree of fermentation. Generally, an FI value between 1 and 1.2 is considered as sufficiently fermented cocoa [37,50,53]; according to the results obtained, it can be indicated that regardless of the use of culture or not acceptable indexes being recorded, this is related to what was indicated by Ooi et al. [54], where there was no variation of IF in cocoa fermented with and without yeast starter culture. In the cocoa from Guadalupe and Tolopampa, an FI indicating adequate fermentation was obtained at 72 h, which was similar to what was obtained by Febrianto and Zhu [15], Ooi et al. [54], and Racine et al. [37]; however, in the cocoa from Copallín, an adequate FI was only obtained at 120 h, which could be due to other factors such as the origin or conditions of the fermentation process such as pH [55]. However, in the cocoa from Guadalupe (SF and SC) and Tolopampa (SF), the FI was higher than 1.6, i.e., (Appendix A), over-fermented, which suggests taking into account the origin of the cocoa under any type of fermentation (spontaneous or with starter culture) [54].

In our study, Aw in SF ranged from 0.89 to 0.96 and in SC from 0.87 to 0.92, as shown in Figure 3, showing slightly lower values, and these results were similar to those obtained by Delgado-Ospina et al. [56]. It can be seen that the Aw showed low values between 72 and 96 h in the fermentations carried out, which, together with IF, internal pH, and bean moisture, can become physicochemical indicators of the degree of fermentation of cocoa [49]. In the SF, the %MC was not very representative; the beans of Guadalupe, Tolopampa, and Copallín started with 29.47%, 32.93%, and 29.93%, ending with 36.19%, 23.39%, and 28.08%. In comparison, in the SC, the results started with 33.10%, 27.91%, and 31.56%, ending with a slight increase to 35.94%, 31.16%, and 34.18%, respectively (Appendix A). These data confirm the findings of Calvo et al. [39], who stated that origin does not affect cocoa moisture.

Acidity is an indicator obtained during bean fermentation [29]; the data recorded indicate that the titratable acidity (TTA) presents stable levels until 24 h, where it begins to increase up to 120 h. Then, it decreases in cocoa from Guadalupe and Tolopampa, while Copallín begins to decrease at 144 h in SF (Appendix A). In SC, high TTA values were recorded at 96 h in cocoa from Tolopampa and Copallín and at 144 h in cocoa from Guadalupe. These behaviors are similar to those reported by Chagas et al. [20], where the highest acidity values occur at the end of the SC due to the presence of *S. Cerevisiae*, which promotes the production of lactic and acetic acid [57,58,59]. Figure 3 proves that while pH decreases, acidity increases [60], and this behavior observed in our study is considered an indicator of good bean fermentation [61].

### 3.3. Antioxidant Profile during Fermentation (SF and SC) in Cocoa Beans

Table 1 and Table 2 show the average values of the main phenolics responsible for the antioxidant profile in cocoa, which were subjected to multivariate analysis (k-means and PCA), as shown in Figure 4 and Figure 5. The polyphenol content (TPC), regardless of the type of fermentation, decreased as the processing time elapsed, registering the lowest value in cocoa from Guadalupe (1.01 ± 0.75 mg (GAE)/g in SF); this reduction in the results is within what is reported in the literature [4,20], which is associated with polyphenol oxidase (PPO) activity, which generates enzymatic browning reactions and the oxidation of phenols diffused in cell fluids [62,63].

It has been reported that within the first 48 h of fermentation, there was a higher level of TPC in cocoa beans (15.5 mg (GAE)/g in Copallín SF) since, in the first days of the process, polyphenols are stored in pigmented cells of the cotyledons [64,65]. In addition, there was an increase in TPC at the end of fermentation, which according to Ho et al. (2014), is not uniform; in our study, at the end of SC fermentation, the TPC content was higher than SF (Table 1 and Table 2), similar to that reported by Chagas et al. [20], suggesting that lower levels of SF would be expected. On the other hand, the increase in TPC could be due to the release of bound phenolic compounds by *S. cerevisiae* [54]. The higher level of TPC in Copallín bean could be due to its origin, which is above 800 masl [66], but further studies are needed.

Mihai et al. [31] indicate that the most abundant polyphenols are anthocyanins, which in our study are significantly affected (*p* < 0.05) by the fermentation time (Table 1 and Table 2). These results are in agreement with those reported by Afoakwa et al. [67], where it is evident that the anthocyanin content is time-dependent. In both types of fermentation, a high anthocyanin content is observed at the beginning, which decreases during the process. These results are similar to that reported by Melo et al. [29] possibly due to the hydrolysis and polymerization of condensed tannins [68], leaching with fermentation exudate [4], and the oxidation by PPO to anthocyanidins [10,69], among others. The decrease in anthocyanins during cocoa fermentation is associated with an increase in temperature (Figure 2) and a decrease in TPC (Table 1 and Table 2) [70,71], which is an indicator of a fermentation process [53,56,68]. In our study, it is observed that SC (Table 2) presented a more significant reduction in anthocyanins around 48 h compared to SF, similar to that obtained by Romanens et al. [72], which was possibly due to the action of *S. cerevisiae* that accelerates the hydrolysis of anthocyanins [73], which remained stable after 120 h. This behavior is similar to that reported by Febrianto and Zhu [15].

The antioxidant activity (AA) and TPC were reduced between 20% and 40% in both fermentations, similar to those reported by Calvo et al. [39], Di Mattia et al. [68], and Suazo et al. [74]. In the first 48 h of fermentation, AA values remained constant, similar to those obtained by Melo et al. [29]. However, the duration of fermentation (*p* < 0.05), together with the maturity at harvest, affects the antioxidant profile of the bean [75,76]. According to DPPH free radical scavenging, the activity in SF and SC was in the range of 57.1–92.4 μmol (TE)/g and 45.7–93.9 μmol (TE)/g, respectively, which were lower values than those reported by Ooi et al. (2020) because they worked with kernels separated from the pulp, demonstrating the contribution of the pulp to AA [77]. This behavior was confirmed by the ABTS method (Table 1 and Table 2), where a 30% reduction concerning the initial value was observed, similar to that reported by Suazo et al. [74] for Trinitario cocoa. Similarly, the AA value by FRAP decreased from 249.0 and 283.0 μmolFe^2+^/100 g to 20.8 and 62.7 μmolFe^2+^/100 g for SF and SC, respectively, which is in agreement with that reported by Melo et al. [29].

In our study, it is clear that the fermentation process affects AA (*p* < 0.05) [76], which is accompanied by a decrease in polyphenols [52]. Regarding the activity by DPPH and ABTS, the values were lower in fermentations with starter culture compared to SF, since yeast species such as *S. cerevisiae*, according to Ooi et al. [54], could influence the enzymatic hydrolysis, generating a low free radical scavenging activity.

Balcázar-Zumaeta et al. [10] and Chagas et al. [20] point out that epicatechin and catechin are flavan-3-ols that contribute to cocoa OA because they are considered bioactive [15] together with proanthocyanidin and anthocyanins. In our study (Table 1 and Table 2), a decrease in epicatechin content was observed in SF and SC, which were 30.4–1.46 mg/g and 24.4–3.11 mg/g, respectively. These values are within the range reported by Balcázar-Zumaeta et al. [10] and Samaniego et al. [78], where epicatechin is generally between 1.17 and 41.73 mg/g, being a monomer with higher bioavailability [53]. Epicatechin is the main monomer in the cocoa beans, which at the beginning of fermentation shows high concentrations in SF (21.7 mg/g) and with starter culture (23.3 mg/g), which are far from previous studies by Melo et al. [29]; these values are reduced by more than 50% after 168 h. Time is a significant factor in its reduction (*p* < 0.05) [29,37,79], which could be due to diffusion outside the cotyledon [10,37,80].

Catechin presented a maximum concentration in SF (0.96 mg/g) and SC (0.93 mg/g), which was the lowest concentration concerning the quantified metabolites; on the other hand, as expected, epicatechin showed higher values [28,80], contrary to what was reported by Melo et al. [29], Samaniego et al. [78] and Septianti et al. [76]. Moreover, it was observed that catechin concentrations were higher at the beginning and gradually decreased during the fermentation process until they completely disappeared by 48 h (SF) and 72 h (SC). The reduction in catechin during fermentation is consistent with what has been reported by Aprotosoaie et al. [79], Melo et al. [29], and Septianti et al. [76], where time (*p* < 0.05) is an influential factor [76].

The concentration of flavan-3-ols decreases during fermentation, which affects the antioxidant capacity [39], which is a characteristic of fermentation [80] possibly influenced by genetic peculiarities (e.g., anatomical characteristics of the bean) [51].

The methylxanthines reported as theobromine and caffeine in fermented beans ranged from 12.1 to 20.7 and 0.24 to 9.96 mg/g, respectively (Table 1 and Table 2), which are values higher than those reported by Quelal-Vásconez et al. [81] and Bustamante et al. [82], where theobromine ranged from 1.53 to 2.4 mg/g and caffeine ranged from 0.15 to 0.41 mg/g. These alkaloids appear during the ripening of the bean, which is contained in pigmented cells [75,83], and they are responsible for the aroma and taste of cocoa [25].

Theobromine increased from 72 to 96 h (SF) and 48 to 72 h (SC). It was also observed that fermentation with starter culture reduced the time to reach a higher theobromine content, which was possibly because the yeast accelerates the embryo’s death to diffuse the content in the tissues [25]. This behavior agrees with the conclusions of M. D. R. Brunetto et al. [84] and Calvo et al. [39], where the increase was detected in the first hours of the process and decreased at 72 h due to the exudation of the beans [39,79], which was due to the process of “biodetheobromination” [10,85].

The presence of caffeine reported in the study is a defense mechanism of cocoa [80]; this methylxanthine, unlike theobromine, had a lower content in the beans, which coincides with that expressed by M. D. R. Brunetto et al. [84] and Septianti et al. [76]. The study showed that the fermentation time affected the caffeine content (*p* < 0.05), with a high content in the first 24 h of the process and a subsequent decrease similar to the behavior reported by Calvo et al. [39] for fresh beans subjected to fermentation. The lowest caffeine content was obtained in Guadalupe cocoa, reaching 0.52 (SF) and 0.24 (SC) mg/g, where it can be inferred that the cultivation conditions influence the chemical composition of the cultivar [75]. In both fermentations, it can be observed that the concentration decreases at the end of the process after having experienced an increase in the first hours; such behavior is consistent with that reported by Calvo et al. [39] and Septianti et al. [76].

### 3.4. Characterization of the Stages during Fermentation (SF and SC) of Cocoa Beans

A multivariate analysis was conducted through the k-means to group the fermentation times (stages) of beans in function of the antioxidant profile. The SC clusters obtained a relationship between the sum of squares between groups and the total sum of squares of 72.7%, superior to SF clusters, which was 66.8%. With these indicators, Figure 4a and Figure 5a suggest that the formation between groups according to the times of each fermentation is reliable.

In SF (Figure 4a,b), the formation of three fermentation stages (clusters) was observed. The first group is characterized by containing the first hours of fermentation (0 h and 24 h, cluster 3) and the highest values of AA, TPC, AC, caffeine, epicatechin, and theobromine; the second group (cluster 2) was conformed by fermentation times from 48 to 72 h, which was characterized by a high content of AA (DPPH) and catechin. Finally, the next group (cluster 1) was formed by fermentation, mostly between 96 and 168 h, presenting lower values of AA, TPC, AC, methylxanthines, and flavan-3-ols. In addition to the k-means analysis, PCA showed that in SF, the sum of the first components (PC1 + PC2) explained 81.6% of the variance of the main data (Figure 4d), which was accompanied by high eigenvalues in each case (6.67 and 0.66, respectively). In the case of the first component, it can be observed that AA, TPC, AC, and flavan-3-ols are the variables with the highest correlation and contribution (4c). In contrast, in the case of methylxanthines, they make the highest contribution to the second component (higher correlation of caffeine).

According to the stages formed, the first one, formed by the times from 0 to 24 h (cluster 3), is characterized by the presentation of greater astringency and bitterness [10], which, according to the data, are associated with a higher content of methylxanthines and flavan-3-ols and accompanied by a high level of TPC, AC and AA; these are similar to those determined by multivariate analysis by Calvo et al. [39] and Febrianto and Zhu [15,53]. In the second fermentation stage (from 48 to 72 h), high concentrations of antioxidant profile variables were observed; these results are similar to those reported by do Carmo et al. [4]; on the other hand, these values were lower than those observed in the first fermentation stage (cluster 3), because during this second stage, embryo death occurs, causing the leaching of cellular compounds responsible for the antioxidant profile [86]. This decrease between stages contributes to the improvement of the sensory properties of cocoa [15,87]. In the third fermentation stage, we observed that the antioxidant profile variables studied are lower than in the first two stages, suggesting that prolonged fermentations may decrease the antioxidant potential in cocoa [15].

In SC, three stages were also obtained (Figure 5a,b), similar to those obtained by Chagas et al. [20]. The first stage (cluster 3, 0 to 48 h) shows the highest AA, TPC, methylxanthines, and epicatechin values. In the second stage (cluster 2), it can be observed that it shows low values of the antioxidant profile and a more significant variation in fermentation times (see Figure 5a). Finally, the third stage (cluster 1) groups mostly at times greater than 120 h, which is characterized by similar concentrations of caffeine and catechin of the first and second stages.

Likewise, the PCA showed that PC1 + PC2 explained 86.0% of the variance of the data, considering the times as observations (5d), and it was noted that PC1 had a higher eigenvalue (6.96). According to the correlation circle (5c), the first component shows a more significant contribution of TPC, AA, and flavan-3ols. At the same time, at the other extreme, we located methylxanthines and AC according to the factor map (5d), which is a behavior similar to SF. The first phase (0 to 48 h) is in agreement with previous studies reported by Chagas et al. [20] and Cheng et al. [88], indicating that it is characterized by high levels of TPC, AA, AC, methylxanthines, and flavan-3ols in response to bacterial activity in the cocoa bean [89]. In addition, we observed that the antioxidant profile in SC remained at higher concentrations for a longer time compared to SF, which was possibly because the yeast facilitates their release (antioxidant compounds) in the bean [20,90]. In both fermentations, prolonged fermentation times decrease the content of the antioxidant profile in SF and SC, demonstrating that it is necessary to reduce the fermentation time [20].

### 3.5. Characterization of the Profile of Volatile Compounds (VCs) Monitored during Fermentation in Cocoa Beans

Figure 6 shows that 56 key compounds were identified in SF. As can be seen in Figure 7a (the plot is from bottom to top; there are three numbers around each node, and the number under each node indicates the rank of the cluster), SF showed discrimination according to VCs in two groups; the main branch on the left grouped the times between 24 and 72 h of fermentation, while the main branch on the right is made up of fermentation times greater than 96 h and the cocoa beans at the beginning of the process (0 h).

During SF, one of the aldehydes in the bean that showed relatively constant concentrations during fermentation was benzene–acetaldehyde (Appendix A), which is similar to that reported by Bastos et al. [91]; likewise, its constant presence during fermentation distinguishes it from other varieties, such as Forastero, where it only appeared at 120 h [92], giving floral and sweet notes [92,93]. In addition, in the case of alcohols, an increase in concentration in the bean was observed after 24 h, which was followed by a slight decrease toward the end of the process; these results are similar to those obtained by Bastos et al. [91]. In this group, the constant presence of 2-pentanol and 2-heptanol stands out, which are compounds intrinsic to the bean that develop during fermentation [7,12] and present fruity and floral notes that are important for aroma and flavor [93,94].

On the other hand, ethanol was identified in cocoa beans (Appendix A), where a higher concentration was reported up to 48 h of fermentation, as it is produced by yeasts in the anaerobic phase [7]. Then, toward the end of fermentation (>120 h), it tends to disappear in the bean, as it is a source to produce acetic acid, as reported in the present study, which is in agreement with what was mentioned by Balcázar-Zumaeta et al. [10] and Qin et al. [94]. A ketone reported in different studies is 2-pentanone [10,12].

Within 48 to 96 h of fermentation, significant concentrations of acetoin were detected (Appendix A); this aldehyde produces a cream and butter odor in SF [12,95]; however, unlike the study by Rodriguez-Campos et al. [96], it was not the aldehyde with the highest concentration in our study. During the same period, a high concentration of 3-methyl-1-butanol was observed, consistent with that reported by Lee et al. [97], which is responsible for producing desirable flavor notes in cocoa [95]. Linalool, a terpene responsible for the fine aroma of cocoa [3] was found during cocoa spontaneous fermentation, similar to that reported by Britto et al. [18], Calvo et al. [39], and Rottiers et al. [12],.

Fifty-five key compounds were identified in the SC (Figure 8); moreover, according to the hierarchical cluster (Figure 7b), there are two main branches; the left one groups the fermentation times of 48 h and 96 h (cocoa beans from Copallín) and 72 h (cocoa beans from Guadalupe). Nevertheless, if we analyze the right main branch, we observe two quite clear divisions according to the times, which group cocoa beans fermented between 96 and 168 h and between 0 and 120 h of fermentation. Among the aldehydes identified, 3-methyl-butanal was detected after 72 h in SC (Appendix A), as reported by Assi-Clair et al. [22], which is considered a key biomarker of aroma in cocoa with malty notes [98]. In addition, similar to the study by Chagas et al. [99], the presence of alcohols in SC (e.g., 2-heptanol, 2-pentanol, 2-nonanol, among others) was identified due to yeast metabolism in the anaerobic phase.

The ester 2-pentanol acetate is a compound with fruity notes present in both fermentations at 48 h, which is similar to that reported by Cevallos-Cevallos et al. [92]; in SC, its concentration remained relatively high until the end of fermentation compared to SF, which can be assumed that the yeast contributed to the formation of the compound (where its highest concentration was between 48 and 96 h). Regarding acids, one of the predominant compounds in SC was acetic acid (Appendix A), which was similar to that reported by Chagas et al. [99] and Sandoval-Lozano et al. [100], which began to increase dramatically between 48 and 72 h of fermentation, a period in which it is associated with a conversion of ethanol to acetic acid, where the yeast *S. cerevisiae* contributes to the oxidation of alcohol [22,99,100]. A higher content of linalool could be observed in SC compared to SF, as reported by Assi-Clair et al. [22]; our study showed high concentrations after 48 h of fermentation similar to Klis et al. [101]; finally, a shorter presence of isobutyl acetate (48 h) was detected in SC, presenting floral, fruity and banana notes [102], as detected in the same type of fermentation in the study of Assi-Clair et al. [22].

However, our study identified undesirable volatile compounds such as 3-methyl-1-butanol, 1-pentanol, 2-methyl-1-propanol, 3-methylbutanoic acid, and acetic acid, which is similar to those reported by Qin et al. [94], who found these volatile compounds after 72 h of fermentation (with rancid, waxy or sulfurous notes). In addition, the presence of 2,4,5-trimethyl-1,3-dioxolane was detected in both fermentations (Appendix A), which is a compound produced by *S. cerevisiae* [102] and previously reported in the study by Valle-Epquín et al. [34] in cocoa from Amazonas as in the research, but in roasted conditions, which may lead to presume that this cocoa contains a significant presence of this yeast, which generates that the addition of the culture does not present many differences in the cocoa bean.

## 4. Conclusions

Our study has allowed us to characterize stages according to the time of spontaneous and starter culture fermentation in cocoa. In SF, the first stage (0 to 24 h) was characterized by a higher content of methylxanthines and flavan-3-ols and a high level of TPC, AC, and AA; subsequently, the second stage was characterized by a high concentration of antioxidant profile variables, but it was lower than in the first stage; finally, in the third stage, the antioxidant profile variables studied were lower than in the first two stages. In the case of SC, the first stage (0 to 48 h) was characterized by higher values of AA, TPC, methylxanthines, and epicatechin; the second stage (72 h to 96 h) was characterized by low values of the antioxidant profile and a more significant variation of fermentation times; finally, the last hours of fermentation (stage 3) were characterized by similar concentrations of caffeine and catechin to the first and second stages.

The study allowed monitoring the concentration of volatile compounds in cocoa beans, highlighting that the leading families reported were acids, alcohols, aldehydes, ketones, esters, hydrocarbons, and furans. In both fermentations, a high concentration of the main aromatic VCs (e.g., acetoin, ethanol, 2-heptanol, 2-pentanol, acetic acid, 3-methyl-1-butanol, among others) can be observed between 48 and 96 h for the cocoa bean. It was also found that there is a higher concentration of alcohols and esters when a starter culture (*S. cerevisiae*) is used, which was possibly because the yeast facilitates the release of compounds. Finally, the study has shown that prolonged fermentation times decrease the antioxidant potential and affect the presence of VCs precursors of aroma and final flavor in cocoa, demonstrating the need to reduce the fermentation time in native Amazonian cocoa.

## Figures and Tables

**Figure 1 foods-12-03291-f001:**
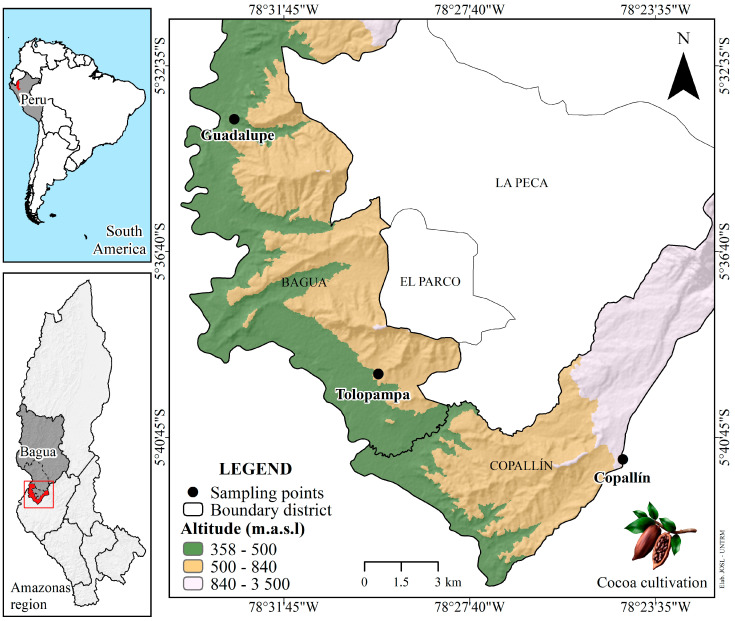
Growth location of cocoa (Amazonas, Peru).

**Figure 2 foods-12-03291-f002:**
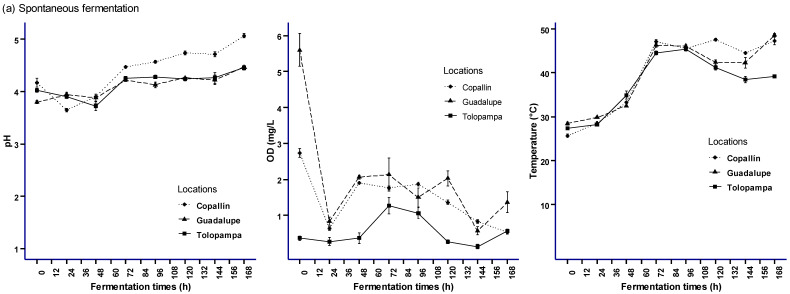
Parameters monitored during (**a**) SF and (**b**) SC of cocoa pulp-bean mass: pH, temperature (°C), and dissolved oxygen (DO). Statistical analyses were performed using RMarkdown software (RStudio, version 2022.07.2+576). The values are presented in the Appendix A.

**Figure 3 foods-12-03291-f003:**
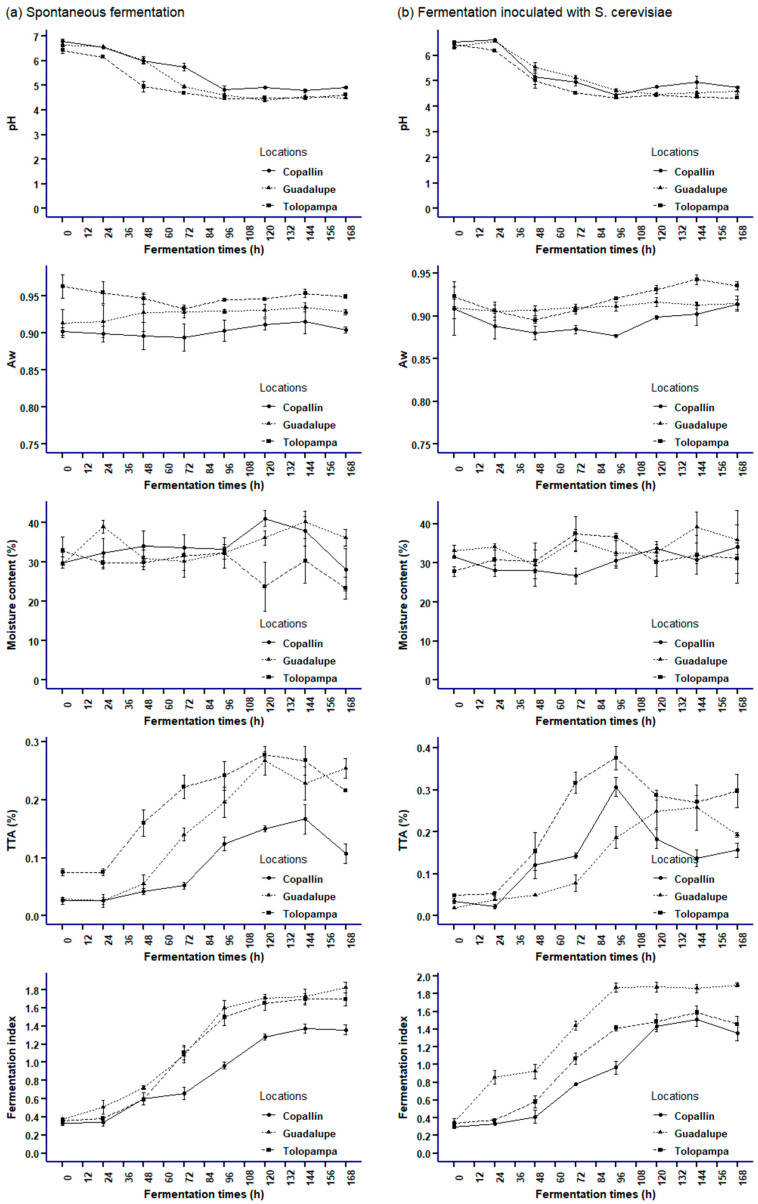
Behavior of physicochemical parameters during (**a**) SF and (**b**) SC of cocoa beans. Statistical analyses were performed using RMarkdown software (RStudio, version 2022.07.2+576). The values are presented in the Appendix A.

**Figure 4 foods-12-03291-f004:**
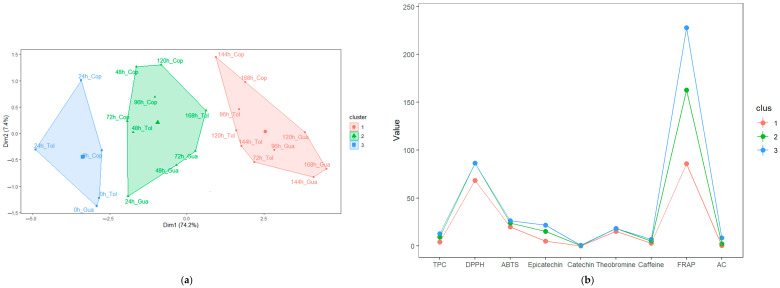
Principal component analysis (PCA) and K-means in cocoa bean spontaneous fermentation: (**a**) K-means clustering shows three stages according to the antioxidant profile variables; (**b**) Mean values of the antioxidant profile variables in each cluster; (**c**) Biplot of principal components analysis of the antioxidant profile variables; (**d**) Principal component analysis (PCA) of the fermentation times based on the contribution of each observation. Antioxidant profile variables were TPC, activity antioxidant (by DPPH, ABTS and FRAP), anthocyanin content (AC), flavan-3-oles epicatechin and catechin), and methylxanthines (theobromine and caffeine). The data used are reported in Table 1. Statistical analyses were performed using RMarkdown software (RStudio, version 2022.07.2+576).

**Figure 5 foods-12-03291-f005:**
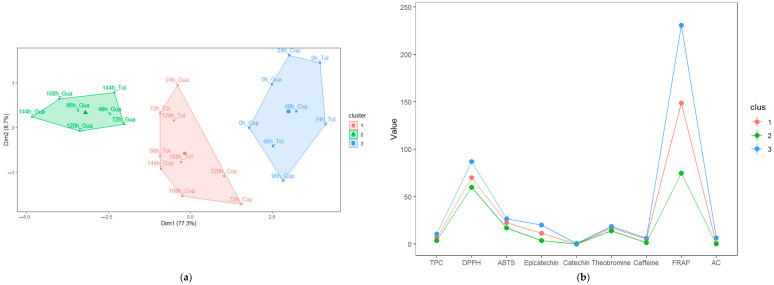
Principal component analysis (PCA) and K-means in cocoa bean fermentation with starter culture: (**a**) K-means clustering shows three stages according to the antioxidant profile variables; (**b**) Mean values of the antioxidant profile variables in each cluster; (**c**) Biplot of principal components analysis of the antioxidant profile variables; (**d**) Principal component analysis (PCA) of the fermentation times based on the contribution of each observation. Antioxidant profile variables were TPC, activity antioxidant (by DPPH, ABTS and FRAP), anthocyanin content (AC), flavan-3-oles epicatechin and catechin), and methylxanthines (theobromine and caffeine). The data used are reported in Table 1. Statistical analyses were performed using RMarkdown software (RStudio, version 2022.07.2+576).

**Figure 6 foods-12-03291-f006:**
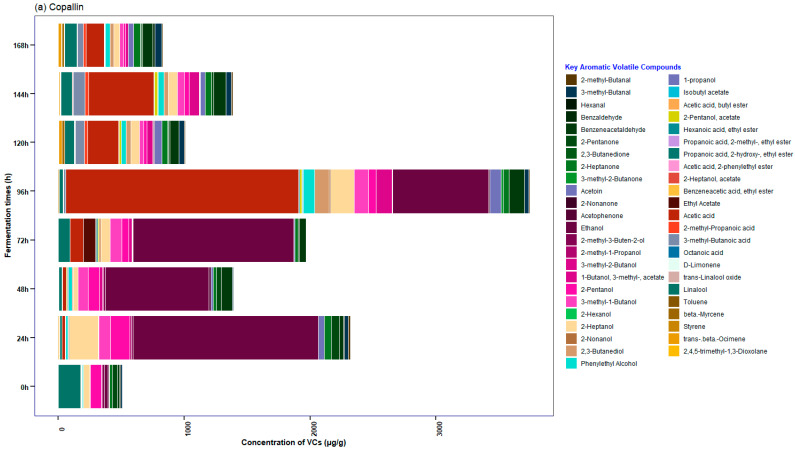
Relative concentrations (μg/g) of aromatic volatile compounds identified by GC-MS during spontaneous fermentation. All experiments were performed for each cocoa beans origins of three populated centers (Copallín, Guadalupe and Tolopampa). Statistical analyses were performed using RMarkdown software (RStudio, version 2022.07.2+576). The data used in the figures are in the Appendix A.

**Figure 7 foods-12-03291-f007:**
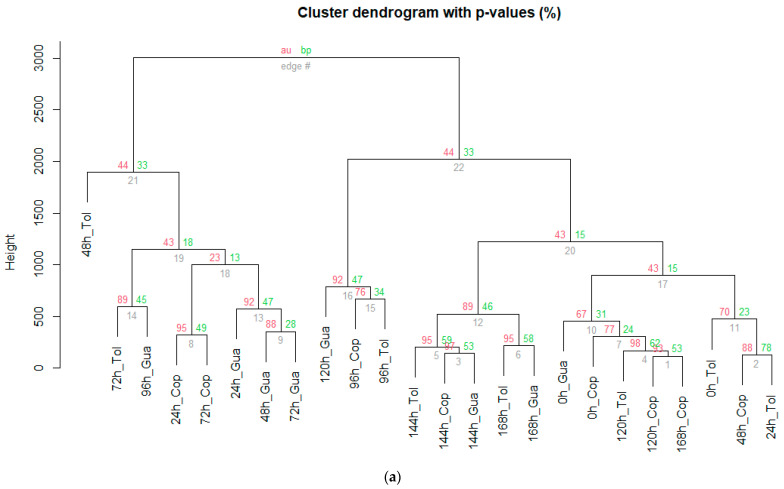
Cluster dendogram of the hierarchical analysis in (**a**) spontaneous fermentation and (**b**) fermentation with starter culture (*S. cerevisiae*). The dissimilarity between the fermentation times was the complete linkage method with Euclidean distance. The two numbers above each node indicate two types of *p*-values calculated via two different bootstrapping algorithms: AU and BP. The number on the left indicates an “approximately unbiased” *p*-value (AU) and is computed by multiscale bootstrap resampling. The number on the right indicates a “bootstrap probability” *p*-value (BP) and is computed by normal bootstrap resampling. The number on the left is a much better assessment of how strongly the cluster is supported by the data. Statistical analyses were performed using RMarkdown software (RStudio, version 2022.07.2+576).

**Figure 8 foods-12-03291-f008:**
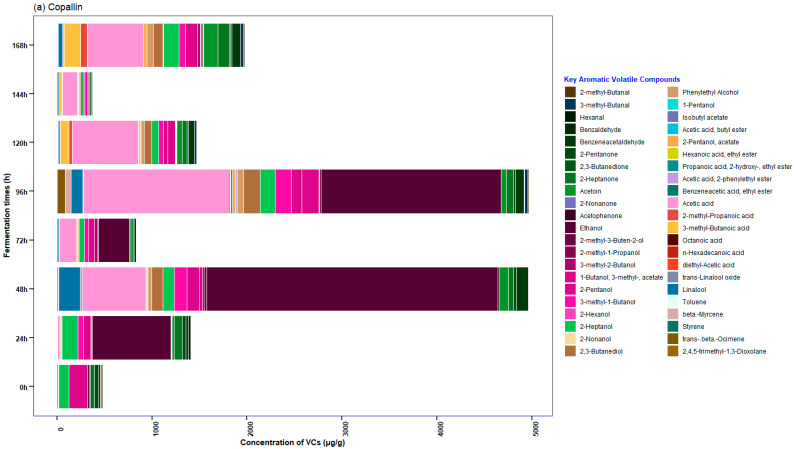
Relative concentrations (μg/g) of aromatic volatile compounds identified by GC-MS during fermentation with starter culture (*S. cerevisiae* All experiments were performed for each cocoa beans origins of three populated centers (Copallín, Guadalupe and Tolopampa). Statistical analyses were performed using RMarkdown software (RStudio, version 2022.07.2+576). The data used in the figures are in the Appendix A.

**Table 1 foods-12-03291-t001:** Average values * of the main phenols during spontaneous fermentation of cocoa beans.

Locations	Fermentation Time (h)	Total phenol Content ^1^	Anthocyanin Content ^2^	Antioxidant Activity			Flavan-3-Ols ^3^		Methylxanthines ^3^	
				DPPH ^4^	ABTS ^4^	FRAP ^5^	Epicatechin	Catechin	Theobromine	Caffeine
Copallín	0	11.70 ^b^ [0.21]	9.49 ^a^ [0.08]	85.50 ^a^ [1.99]	26.90 ^a^ [0.14]	226.00 ^a^ [3. 21]	14.70 ^ab^ [1.54]	0.44 ^a^ [0.09]	17.90 ^a^ [0.75]	7.09 ^bc^ [1.05]
	24	15.50 ^a^ [0.34]	7.82 ^b^ [0.06]	81.50 ^ab^ [0.14]	27.10 ^a^ [0.32]	184.00 ^bc^ [11.50]	21.70 ^a^ [4.07]	0.44 ^a^ [0.14]	18.70 ^a^ [1.21]	9.96 ^a^ [0.92]
	48	13.00 ^b^ [0.84]	3.18 ^d^ [0.02]	79.80 ^ab^ [13.5]	26.70 ^a^ [0.64]	149.00 ^d^ [13.30]	16.70 ^ab^ [4.82]	0.19 ^ab^ [0.05]	18.80 ^a^ [0.13]	7.36 ^bc^ [1.07]
	72	12.20 ^b^ [0.31]	3.61 ^c^ [0.07]	92.40 ^a^ [0. 83]	25.30 ^a^ [0.50]	198.00 ^ab^ [7.60]	20.00 ^a^ [2.02]	0.16 ^ab^ [0.14]	18.50 ^a^ [1.02]	5.36 ^cd^ [0.27]
	96	9.62 ^c^ [0.30]	2.00 ^e^ [0.07]	88.40 ^a^ [6. 21]	23.00 ^b^ [0.92]	170.00 ^bcd^ [7.90]	15.60 ^ab^ [0.71]	0.20 ^ab^ [0.03]	19.10 ^a^ [0.08]	5.56 ^c^ [0.36]
	120	8.56 ^c^ [0.19]	1.29 ^f^ [0.01]	83.50 ^a^ [2. 39]	22.90 ^bc^ [0.90]	160.00 ^cd^ [12.90]	12.10 ^bc^ [0.25]	0.32 ^ab^ [0.05]	18.00 ^a^ [0.15]	8.85 ^ab^ [1.27]
	144	6.39 ^d^ [0.88]	0.74 ^g^ [0.04]	66.30 ^bc^ [4.78]	21.10 ^bc^ [0.94]	116.00 ^e^ [8.95]	6.42 ^c^ [1.20]	0.20 ^ab^ [0.28]	18.30 ^a^ [0.47]	6.35 ^c^ [0.10]
	168	5.25 ^d^ [0.41]	0.80 ^g^ [0.08]	62.40 ^c^ [6.19]	20.90 ^c^ [0.90]	104.00 ^e^ [12.90]	5.42 ^c^ [2.57]	0.00 ^b^ [0.00]	18.20 ^a^ [0.25]	3.16 ^d^ [0.33]
Guadalupe	0	10.90 ^a^ [0.34]	6.13 ^a^ [0.08]	83.40 ^ab^ [4.65]	26.90 ^a^ [0.81]	218.00 ^a^ [8.99]	21.70 ^a^ [0.11]	0.96 ^a^ [0.05]	18.40 ^a^ [0.39]	3.94 ^a^ [0.72]
	24	9.32 ^a^ [0.14]	4.63 ^b^ [0.08]	90.40 ^a^ [5.41]	25.20 ^b^ [0.37]	224.00 ^a^ [9.21]	19.00 ^a^ [1.86]	0.53 ^b^ [0.08]	18.20 ^a^ [0.70]	2.78 ^b^ [0.39]
	48	6.93 ^b^ [0.33]	2.43 ^c^ [0.08]	87.90 ^ab^ [11.0]	26.80 ^a^ [0.17]	144.00 ^b^ [4.78]	9.30 ^b^ [3.49]	0.36 ^b^ [0.29]	17.70 ^a^ [0.43]	2.84 ^b^ [0.15]
	72	7.13 ^b^ [1.12]	0.71 ^d^ [0.01]	83.60 ^ab^ [5.66]	25.90 ^ab^ [0.62]	110.00 ^c^ [6.75]	12.30 ^b^ [2.27]	0.26 ^bc^ [0.10]	17.40 ^a^ [0.16]	2.35 ^bc^ [0.07]
	96	4.05 ^c^ [0.30]	0.25 ^e^ [0.02]	74.80 ^bc^ [3.68]	21.40 ^c^ [0.21]	51.60 ^d^ [6.83]	2.76 ^c^ [0.57]	0.00 ^c^ [0.00]	14.90 ^b^ [0.06]	1.61 ^cd^ [0.14]
	120	3.82 ^cd^ [0.62]	0.22 ^e^ [0.00]	57.10 ^d^ [0.66]	16.40 ^e^ [0.68]	51.00 ^d^ [2.51]	2.90 ^c^ [1.41]	0.00 ^c^ [0.00]	14.80 ^b^ [1.07]	1.20 ^de^ [0.21]
	144	2.42 ^de^ [0.12]	0.18 ^e^ [0.08]	67.40 ^cd^ [1.81]	15.90 ^e^ [0.14]	49.80 ^d^ [7.40]	2.82 ^c^ [0.59]	0.00 ^c^ [0.00]	13.00 ^b^ [0.24]	0.64 ^e^ [0.09]
	168	1.01 ^e^ [0.75]	0.11 ^e^ [0.04]	59.30 ^d^ [0.90]	17.90 ^d^ [0.41]	20.80 ^e^ [4.36]	1.46 ^c^ [0.73]	0.00 ^c^ [0.00]	13.00 ^b^ [1.63]	0.52 ^e^ [0.19]
Tolopampa	0	11.70 ^ab^ [0.63]	10.30 ^a^ [0.08]	91.70 ^a^ [0. 52]	25.40 ^a^ [1.34]	249.00 ^a^ [1.76]	20.80 ^b^ [1.83]	0.35 ^bc^ [0.10]	17.30 ^bc^ [0.44]	4.90 ^bc^ [0.04]
	24	14.20 ^a^ [0.58]	9.42 ^b^ [0.08]	89.10 ^ab^ [1.26]	26.10 ^a^ [1.11]	263.00 ^a^ [0.77]	30.40 ^a^ [1.06]	0.86 ^a^ [0.09]	19.10 ^a^ [0.39]	9.21 ^a^ [0.73]
	48	9.96 ^bc^ [1.83]	3.72 ^c^ [0.06]	90.40 ^ab^ [2.53]	22.00 ^b^ [0.58]	199.00 ^b^ [10.80]	20.30 ^b^ [0.59]	0.40 ^b^ [0.00]	18.20 ^ab^ [0.48]	6.14 ^b^ [0.63]
	72	3.64 ^d^ [0.81]	1.14 ^d^ [0.06]	79.20 ^ab^ [0.50]	21.10 ^bc^ [0.37]	115.00 ^c^ [6.64]	5.02 ^d^ [4.09]	0.00 ^d^ [0.00]	14.20 ^e^ [0.49]	2.92 ^c^ [0.21]
	96	4.30 ^d^ [1.45]	0.79 ^e^ [0.04]	78.30 ^b^ [3.75]	19.70 ^cd^ [0.65]	117.00 ^c^ [0.79]	6.50 ^cd^ [0.40]	0.00 ^d^ [0.00]	16.10 ^cd^ [0.06]	4.81 ^bc^ [1.55]
	120	5.36 ^d^ [0.38]	0.77 ^e^ [0.02]	58.80 ^c^ [6.30]	25.70 ^a^ [0.19]	124.00 ^c^ [13.30]	10.10 ^c^ [0.58]	0.09 ^d^ [0.00]	15.30 ^de^ [0.74]	3.34 ^c^ [0.78]
	144	3.98 ^d^ [0.94]	0.47 ^f^ [0.01]	79.70 ^ab^ [5.07]	19.10 ^cd^ [0.93]	110.00 ^c^ [12.40]	8.60 ^cd^ [0.20]	0.16 ^cd^ [0.15]	15.40 ^de^ [0.04]	3.62 ^c^ [0.03]
	168	6.50 ^cd^ [2.15]	0.40 ^f^ [0.03]	79.80 ^ab^ [8.23]	18.00 ^d^ [0.17]	112.00 ^c^ [0.82]	11.00 ^c^ [1.59]	0.43 ^b^ [0.03]	18.20 ^ab^ [0.90]	4.54 ^bc^ [0.25]

* Means and standard deviation in brackets, with different lowercase letters in the same columns (fermentation time) indicating statistically different (Tukey test, *p* ≤ 0.05). ^1^ TPC expressed in mg (GAE)/g of sample. ^2^ AC expressed in mg of cyanidin-3-glucoside/100 g of sample. ^3^ Flavan-3-ols (epicatechin and catechin) and methylxanthines (theobromine and caffeine) were expressed in mg of the metabolite/g sample. ^4^ Antioxidant activity by DPPH and ABTS techniques expressed in µmol TE/g sample. ^5^ Antioxidant activity by FRAP technique expressed in μmol Fe^2+^/100 g sample.

**Table 2 foods-12-03291-t002:** Average values * of the main phenols during fermentation with a starter culture *(S. cerevisiae)* of cocoa beans.

Locations	Fermentation Time (h)	Total Phenol Content ^1^	Anthocyanin Content ^2^	Antioxidant Activity			Flavan-3-Ols ^3^		Methylxanthines ^3^	
				DPPH ^4^	ABTS ^4^	FRAP ^5^	Epicatechin	Catechin	Theobromine	Caffeine
Copallín	0	6.97 ^de^ [0.18]	6.90 ^c^ [0.08]	59.90 ^b^ [4.35]	27.80 ^a^ [0.11]	242.00 ^a^ [10.40]	16.00 ^bcd^ [3.11]	0.61 ^ab^ [0.28]	17.60 ^b^ [0.29]	6.30 ^ab^ [0.82]
	24	14.40 ^a^ [0.25]	9.05 ^a^ [0.05]	89.70 ^ab^ [0.36]	27.60 ^a^ [0.31]	258.00 ^a^ [1.01]	14.50 ^cd^ [3.41]	0.16 ^c^ [0.02]	17.50 ^b^ [2.26]	6.81 ^ab^ [1.14]
	48	11.70 ^ab^ [0.56]	7.10 ^b^ [0.07]	93.90 ^a^ [0.52]	25.70 ^a^ [0.73]	235.00 ^a^ [11.10]	20.80 ^ab^ [0.40]	0.31 ^bc^ [0.09]	20.70 ^a^ [0.16]	6.54 ^ab^ [0.41]
	72	7.90 ^cde^ [0.14]	1.85 ^d^ [0.08]	74.60 ^ab^ [2.19]	25.30 ^a^ [1.92]	128.00 ^c^ [2.30]	21.80 ^a^ [1.21]	0.43 ^abc^ [0.02]	19.70 ^ab^ [0.09]	8.04 ^a^ [0.51]
	96	10.50 ^bc^ [0.16]	1.83 ^d^ [0.06]	92.40 ^ab^ [1.7]	26.70 ^a^ [0.77]	171.00 ^b^ [11.20]	19.60 ^abc^ [0.62]	0.70 ^a^ [0.06]	19.80 ^ab^ [0.37]	8.42 ^a^ [0.74]
	120	9.54 ^bcd^ [2.54]	0.78 ^e^ [0.07]	67.30 ^ab^ [24.4]	25.60 ^a^ [0.11]	160.00 ^b^ [12.50]	16.80 ^abcd^ [1.30]	0.42 ^abc^ [0.16]	18.00 ^b^ [0.17]	6.83 ^ab^ [1.20]
	144	6.03 ^e^ [0.28]	0.93 ^e^ [0.06]	59.20 ^b^ [8.32]	21.10 ^b^ [0.61]	134.00 ^c^ [7.90]	9.16 ^e^ [0.17]	0.13 ^c^ [0.09]	18.50 ^ab^ [0.15]	4.64 ^b^ [0.08]
	168	5.71 ^e^ [1.72]	0.85 ^e^ [0.01]	60.70 ^ab^ [20.8]	20.90 ^b^ [1.52]	147.00 ^bc^ [3.93]	12.00 ^de^ [1.37]	0.27 ^bc^ [0.13]	19.00 ^ab^ [0.26]	6.65 ^ab^ [0.62]
Guadalupe	0	11.40 ^a^ [1.19]	5.80 ^a^ [0.02]	86.10 ^a^ [2.21]	25.80 ^a^ [0.32]	210.00 ^a^ [6.80]	20.90 ^a^ [1.27]	0.84 ^a^ [0.09]	17.40 ^a^ [0.14]	3.2 ^a^ [1.42]
	24	8.03 ^b^ [0.18]	0.81 ^b^ [0.05]	83.30 ^ab^ [4.07]	26.30 ^a^ [1.34]	173.00 ^b^ [7.94]	6.49 ^b^ [2.59]	0.06 ^b^ [0.06]	15.50 ^abc^ [0.49]	2.75 ^ab^ [0.39]
	48	4.77 ^cd^ [0.64]	0.59 ^c^ [0.01]	64.20 ^c^ [4.08]	20.70 ^bc^ [1.95]	84.40 ^c^ [12.10]	5.07 ^bc^ [1.39]	0.00 ^b^ [0.00]	14.60 ^bcd^ [1.31]	2.13 ^abc^ [0.87]
	72	5.80 ^bc^ [1.46]	0.72 ^b^ [0.01]	67.00 ^bc^ [6.24]	21.50 ^b^ [1.52]	83.10 ^cd^ [11.00]	3.23 ^bc^ [1.98]	0.00 ^b^ [0.00]	15.90 ^ab^ [0.07]	2.87 ^a^ [0.70]
	96	4.04 ^cd^ [0.80]	0.43 ^d^ [0.03]	54.40 ^cd^ [11.9]	17.80 ^cd^ [0.39]	67.20 ^cde^ [11.70]	3.39 ^bc^ [2.60]	0.00 ^b^ [0.00]	13.80 ^cde^ [0.20]	0.91 ^bc^ [0.26]
	120	3.68 ^cde^ [0.66]	0.2 ^e^ [0.01]	57.70 ^cd^ [8.69]	15.20 ^de^ [0.09]	56.00 ^de^ [8.43]	4.54 ^bc^ [1.43]	0.00 ^b^ [0.00]	15.00 ^bc^ [1.43]	1.54 ^abc^ [0.28]
	144	1.50 ^e^ [0.09]	0.15 ^e^ [0.05]	45.70 ^d^ [1.26]	12.10 ^f^ [0.53]	41.10 ^e^ [9.40]	1.29 ^c^ [0.02]	0.00 ^b^ [0.00]	12.80 ^de^ [0.12]	0.24 ^c^ [0.02]
	168	2.52 ^de^ [0.03]	0.16 ^e^ [0.04]	60.00 ^cd^ [2.13]	14.70 ^ef^ [0.76]	62.70 ^cde^ [11.10]	3.11 ^bc^ [0.12]	0.00 ^b^ [0.00]	12.10 ^e^ [0.07]	0.96 ^bc^ [0.07]
Tolopampa	0	12.10 ^a^ [0.20]	9.95 ^a^ [0.05]	92.10 ^a^ [0.99]	28.10 ^a^ [0.09]	283.00 ^a^ [10.70]	23.30 ^a^ [0.29]	0.62 ^ab^ [0.11]	18.60 ^a^ [0.40]	5.31 ^bc^ [0.30]
	24	10.20 ^b^ [0.95]	9.32 ^b^ [0.05]	91.60 ^a^ [0.97]	26.60 ^ab^ [1.10]	227.00 ^b^ [7.69]	24.40 ^a^ [1.37]	0.93 ^a^ [0.26]	19.60 ^a^ [0.74]	7.94 ^a^ [0.10]
	48	9.23 ^b^ [0.24]	2.77 ^c^ [0.08]	89.90 ^a^ [1.26]	25.50 ^b^ [0.86]	220.00 ^b^ [11.80]	22.00 ^a^ [0.85]	0.60 ^b^ [0.06]	19.20 ^a^ [0.28]	6.26 ^b^ [0.61]
	72	5.84 ^cd^ [0.74]	1.13 ^d^ [0.08]	75.20 ^b^ [2.88]	21.00 ^cd^ [0.92]	145.00 ^cd^ [13.20]	10.10 ^b^ [0.33]	0.28 ^c^ [0.05]	14.80 ^cd^ [0.45]	3.67 ^de^ [0.33]
	96	5.56 ^cd^ [1.00]	0.85 ^e^ [0.04]	64.80 ^c^ [4.26]	21.00 ^cd^ [0.12]	137.00 ^cd^ [10.80]	9.42 ^bc^ [1.86]	0.19 ^c^ [0.08]	17.00 ^b^ [0.13]	4.92 ^bcd^ [0.25]
	120	6.38 ^c^ [0.57]	0.80 ^e^ [0.04]	78.80 ^b^ [0.21]	22.70 ^c^ [1.43]	165.00 ^c^ [11.80]	8.00 ^bc^ [0.54]	0.19 ^c^ [0.08]	16.20 ^bc^ [0.29]	4.24 ^cd^ [0.77]
	144	4.05 ^d^ [0.60]	0.76 ^e^ [0.03]	71.10 ^bc^ [6.76]	19.20 ^d^ [0.26]	131.00 ^d^ [5.94]	6.17 ^c^ [1.12]	0.00 ^c^ [0.00]	13.60 ^d^ [0.89]	2.10 ^e^ [0.11]
	168	7.09 ^c^ [0.10]	0.75 ^e^ [0.01]	69.60 ^bc^ [3.14]	21.80 ^c^ [1.21]	148.00 ^cd^ [13.50]	11.30 ^b^ [2.72]	0.15 ^c^ [0.01]	17.10 ^b^ [0.51]	6.38 ^ab^ [1.20]

* Means and standard deviation in brackets, with different lowercased letters in the same columns (fermentation time) indicating statistically different (Tukey test, *p* ≤ 0.05). ^1^ TPC expressed in mg (GAE)/g of sample. ^2^ AC expressed in mg cyanidin-3-glucoside/100 g sample. ^3^ Flavan-3-ols (epicatechin and catechin) and methylxanthines (theobromine and caffeine) was expressed in mg of the metabolite/g sample. ^4^ Antioxidant activity by DPPH and ABTS techniques expressed in µmol TE/g sample. ^5^ Antioxidant activity by FRAP technique expressed in μmol Fe^2+^/100 g sample.

## Data Availability

The data presented in this study are available in Appendix A.

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
