# Peer review of "Reduction in the Cocoa Spontaneous and Starter Culture Fermentation Time Based on the Antioxidant Profile Characterization"

_foods, 2023, doi:10.3390/foods12173291_

Round 1
Reviewer 1 Report
Foods-2574390
The manuscript describes the cocoa’s fermentation characteristics, using row material from diverse origins and spontaneous or starter culture-assisted processes. The information provided by the work is abundant, and the results are reliable, thanks to the equipment used and the personal qualification. However, the presentation is relatively poor. Some graphics do not have the image impact necessary to transmit the information they contain. Also, the English require revision. Some terms could be substituted with more appropriate others, and several punctuation and sentence connexions could be improved. Additionally, although the use of ± sd (or se) is fairly common, it has no statistical meaning. Since the measurement dispersion could be a piece of interesting information, it can be provided in brackets. Finally, more detailed information on equipment suppliers and statistical programs would also be acknowledged. This reviewer encourages authors to revise the manuscript to offer readers a more attractive reading.
Title. Please, rephrase to clarify.
L26-27. Please, rephrase for clarity.
L32 formation?
L58-60. Too long sentence and complex.
L64. This can improve the quality….?
L93. Please identify better the origin of reagents and culture media (yeast)
L108. Fig. 1 is focused on growth locations.
L117. Please, concrete process identification.
L143. Methanol:HCl. Proportion?
L143-147. City?
L152 residues?
L157,161, 168. City? Please revise other cases.
L212-221. Some sentences of the paragraph require corrections and clarification.
L233-235. The sentence is rather vague.
L255-256. RStudio is a framework. Detail the R program used and the R version.
L259. Figure 1 refers to the growing areas.
Table 1,2,. Identify the figures following the averages. Also, the use of ±has not statistical meaning in this context. In the case of sd or se, just include it in a bracket to indicate the measures of dispersion
L281. Require correct Fig identification.
Fig 2,3. Requires improvement. The overall aspect is poor.
Fig 4,5. Identification of the program used for graphs is necessary
Fig 6 and 7 are illegible.
Fig legends. In general, figure legends are insufficient. Particularly, Fig 8 includes numerous information that requires explanation.
Supplementary information.
Tables require including dispersion measure (sd or se) in brackets when appropriate.
Several terms could have more appropriate words. Syntaxis and expressions could be improved in various sections of the manuscript
Author Response
Dear Reviewer, we have considered all the suggestions that were made for you. Thank you for your support, below is a detailed response to each suggestion.

Reviewer 2 Report
The manuscript addresses the topic related to the fermentation duration and its effect on cocoa quality, especially its antioxidant properties, taking into account the use of starter culture.
Several other authors have reported the importance of the fermentation time and use of starter culture for cocoa in different other countries. The manuscript fill the gap in relation to Peru geo-climatic conditions and its cocoa varieties, which brings a certain level of novelty.
Generally, the employed methodology is appropriate, but some concerns are related to how representative is the size of the sampling (e.g. 50g of cocoa beans for a 40 kg batch).
The conclusions are supported by the results and addresses well the initial questions.
Several punctual comments/issues to be fixed are imeized below.
Line 56-60: too long phrase, try to split and make easy to read sentences/phrases
Line 72: replace “We find…” with “There were reported…”
Line 79: replace “by” with “of”
Line 79-81: the phrase is unclear; please rephrase and make clear the cited authors’ findings/results
Line 84: replace “characterizing” with “to characterize”
Line 91: please be more specific on what it means “low, medium, and high”
Line 104: please specify the level (CFU/mL) and % of the inoculum
Figure 1 is not cited and correlated to its explanation in the main text
Line 123: your daily sample was 50g of cocoa beans; is this representative for a 40 kg batch? Is this a usual procedure for sampling?
Line 230: change “we worked”; try to do not use the second person (we) and make it impersonal
Line 259: maybe figure 2 instead of figure 1
Line 266: maybe Table 1 and Table 2 instead of Table 2 and Table 3. Please describe if there is a correlation in this case
Line 281: sure figure 1? Maybe 2…Please check the entire manuscript and apply changes for the cited figures and table, according to the text content
Line 302: there is no need to put in Italic de word lactic and acetobacter (only if you use the scientific nomenclature)
Minor editing is required.
Author Response

(The authors gave the same response as above.)

Round 2
Reviewer 2 Report
No further comments. All issues were addressed.